# Study on the Cellular Anti-Inflammatory Effect of Torularhodin Produced by *Sporidiobolus pararoseus* ZQHL Isolated from Vinegar Fungus

**DOI:** 10.3390/molecules28031436

**Published:** 2023-02-02

**Authors:** Chang Liu, Mei Han, Fuqiang Lv, Yaobin Gao, Xiaoyun Wang, Xujiao Zhang, Yahui Guo, Yuliang Cheng, He Qian

**Affiliations:** 1State Key Laboratory of Food Science and Technology, School of Food Science and Technology, Jiangnan University, No. 1800 Lihu Avenue, Wuxi 214122, China; 2Department of Food Science, Shanghai Business School, Shanghai 200235, China; 3Jiangsu Hengshun Vinegar-Industry Co., Ltd., No. 66 Hengshun Road, Zhenjiang 212143, China; 4Shanxi Mature Vinegar Group Co., Ltd., No. 26 Madaopo, Xinghua District, Taiyuan 030013, China; 5Shanxi Zilin Vinegar Industry Co., Ltd., No. 550 Gaohua Duan, Taimao Road, Taiyuan 030100, China

**Keywords:** vinegar, *Sporidiobolus pararoseus*, torularhodin, antioxidant, anti-inflammatory

## Abstract

The red stretcher bacterium *Sporidiobolus pararoseus* is a high producer of carotenoids such as torularhodin, but its presence in vinegar has not been detected. Moreover, torularhodin has several biological activities, but its effect on the LPS-induced RAW 264.7 inflammatory cell model has also yet to be elucidated. In this study, *S. pararoseus* was identified in different vinegar samples from China by ITS sequencing. Meanwhile, one of the strains was deeply resolved by whole genome sequencing and functional annotation and named *S. pararoseus* ZQHL. Subsequently, the antioxidant effect of the fungal carotenoid torularhodin was investigated using in vitro DPPH, ABTS, and cellular models. Finally, LPS-induced RAW 264.7 cells were used as an inflammation model to assess torularhodin’s protective effect on inflammatory cells and to determine whether the TLR4 pathway is associated with this process. The results indicate that torularhodin has good free radical scavenging ability in vitro and can contribute to cell viability. More importantly, torularhodin alleviated LPS-induced cellular inflammatory damage and reduced the expression of inflammatory factors such as TLR4, MyD88, and TNF-a. The mechanism may attenuate the cellular inflammatory response by inhibiting the TLR4 inflammatory pathway. In conclusion, torularhodin produced by *S. pararoseus* fungi in vinegar samples significantly scavenged free radicals in vitro and alleviated RAW 264.7 cellular inflammation by modulating the TLR4 pathway.

## 1. Introduction

Microbial carotenoids are produced by oil-producing microorganisms such as yeast and algae [1]. Microbial carotenoids offer several advantages over carotenoid products of plant origin, including not competing with food production for arable land, being unaffected by variations in natural conditions such as climate, and using industrial waste or other cheap raw materials as a source of carbon [2]. Currently, yeast is widely considered the most suitable fungus for producing microbial carotenoids. It is also widely used in food, agriculture, medicine, and biofuels [3]. Stretcherella yeasts are found in a wide range of natural sources. In addition to the marine environment, they inhabit terrestrial environments, including soils, fruit and vegetable surfaces, and fermentation plants [4,5]. Some more comprehensively studied *stretcherella* yeasts are *Yarrowia lipolytica*, *Rhodotorula glutinis,* and *Xanthophyllomyces dendrorhous* [6]. The search for yeast strains of safer origin with specific metabolites has become an essential task for many research centers worldwide. *Sporidiobolus pararoseus* is a parthenogenic aerobic tamer yeast and due to having the advantages of simple nutritional requirements, short fermentation cycles, a single mode of reproduction, and high-density culture, its industrial application looks very promising [7].

*S. pararoseus* has been found to produce many functional metabolites of great importance in the contemporary food, feed, and pharmaceutical industries [8]. According to our previous studies, *S. pararoseus* can produce many carotenoids, such as β-carotene, γ-carotene, torulene, and torularhodin [9]. Of these, torularhodin is a unique carotenoid produced by *S. pararoseus*, whose structure differs from that of β-carotene only in the presence of an additional carboxyl group. Torularhodin was found to have significant anticancer effects in vitro and in vivo and to reduce blood lipids in mice fed a high-fat diet [10]. Besides, torularhodin effectively ameliorated oxidative stress, and neuroinflammation, possibly through the Nrf2/NF-κB signaling pathways [11]. Torularhodin also reduced d-galactose-induced liver oxidation via the promotion of the Nrf2/HO-1 pathways in vivo [12]. However, more research is needed on its bioactivity, healthy function, and especially its anti-inflammatory and antioxidant effects in vitro and in vivo [13]. The TLR4 inflammatory pathway has been reported as a representative pathway for inflammatory damage [14]. When the TLR pathway is activated, its bridging molecule MyD88 alters the downstream signaling of TLR4 to induce the production of various inflammatory cytokines, such as interleukin (IL-1β) and tumor necrosis factor-a (TNF-a) [15]. These pro-inflammatory cytokines continue to mediate inflammation and worsen organismal damage. Drugs, while beneficial in inhibiting the progression of inflammation, may, however, produce unwanted side effects. Therefore, it is crucial to alleviate the inflammatory response through dietary supplementation with natural products.

In this study, a strain of *S. pararoseus* as a carotenoid-producing fungus from different Chinese vinegar factories was screened and identified as *S. pararoseus* ZQHL. We then triple-sequenced the fungus to resolve its whole genome and functional annotation. The ability of torularhodin, a specific carotenoid produced by *S. pararoseus*, to scavenge DPPH and ABTS in vitro and to regulate cellular activity was also investigated. In addition, the mechanism of torularhodin’s amelioration of LPS-induced inflammation via the TLR4 pathway in RAW 264.7 cells was investigated. The results of this study will contribute to a better understanding of the origin of *S. pararoseus* and the potential application implications of its product, torularhodin.

## 2. Results and Discussion

### 2.1. Origin and Distribution of Sporidiobolus pararoseus

Animals cannot synthesize carotenoids directly on their own and must obtain them from food. Microbial production of carotenoids has unique economic advantages and industrial application prospects compared to the extraction of carotenoid pigments from plants and vegetables [16]. Therefore, source screening of carotenoid-producing microbial strains is also important. As shown in Figure 1A, we collected samples from six edible vinegar plants across China for ITS sequencing analysis. At the species level, five vinegar plants contained *S. pararoseus* fungi in their samples. Of note, two vinegar plants, a and f, had a high relative abundance of *S. pararoseus* at 0.5% (Figure 1B,C,H). Finally, isolating and purifying single colonies, a strain of lock-throw yeast was screened and sent to the Chinese Typical Microbial Conservation Centre for conservation (no. CCTCC NO: M 2021729) and named *Sporidiobolus pararoseus* ZQHL (Figure 2A).

### 2.2. Genome-Wide and Functional Annotation of Sporidiobolus pararoseus ZQHL

We sequenced the whole genome of the screened *S. pararoseus* ZQHL fungus with a GC% content of 46.96% and performed GO enrichment analysis and functional description of the predicted genes. The number of related genes included in each GO entry was counted, and FDR (q value) values less than 0.05 were considered as enrichment entries. Among the biological processes (Figure 2B), the cellular process, metabolic process, and single-organism process were the pathways with more enriched genes. In addition, KOG is an NCBI-based annotation system for direct homology relationships, and KOG targets eukaryotes. By comparing the protein sequences obtained from gene prediction with the NCBI CDD database, the specific functional information of the genes is predicted, and the functional classification statistics are made and then plotted to understand the functional distribution of the genes in the species at a macro level. In the KOG functional annotation (Figure 2C), the functions of *S. pararoseus* ZQHL fungus are mainly focused on K: Transcription, O: Posttranslational modification, protein turnover, chaperones, T: Signal Whole-genome sequencing, and functional annotation will be of great theoretical value for the subsequent analysis of the systematic application of *S. pararoseus* ZQHL. The demand for carotenoids of natural origin has been proliferating and is expected to outpace the growth of the global market in the coming years [17]. Sporidiobolus in this study represents a unique tamer yeast lineage with high production of multiple carotenoids [18]. Therefore, it is essential to continue to explore its biological value aligned to industrial applications based on whole genome results.

### 2.3. Antioxidant Capacity and Cytotoxicity of Torularhodin in Sporidiobolus pararoseus ZQHL

It is generally accepted that the structure of a compound is closely related to its biological activity and pharmacological effects. Phytochemical carotenoids have been shown to have significant antioxidant functions [19]. Many carotenoids have been widely used as precursors to Vitamin A in food, pharmaceuticals, and as tumor suppressors [20]. Studies have shown that Vitamin A has significant antioxidant effects in scavenging free radicals [21]. Torularhodin is a carotenoid and can be classified as a precursor of Vitamin A. Its structure differs from that of β-carotene by the presence of an additional carboxyl group. It can be assumed that the antioxidant function of torularhodin has a theoretical basis in chemical structure.

In in vitro antioxidant assays, the ABTS method was used to evaluate the free radical scavenging capacity. The DPPH method has also been widely used to test the antioxidant capacity of bioactive compounds due to its hydrogen supply capacity [22]. As shown in Figure 3A,B, torularhodin exhibited better free radical scavenging activity, and both had R^2^ values of 0.98. IC_50_ values were 1.96 μM in ABTS and 9.38 μM in DPPH, and the results showed a dose dependence. These results suggest that torularhodin has free radical scavenging and antioxidant activity. This is in agreement with our previous findings [12]. It has also been shown that carotenoids derived from different sources of microorganisms possess different antioxidant activities and also correlate with carotenoid content [23]. The in vitro free radical scavenging results of torularhodin are in agreement with the presumed chemical structure. Torularhodin is a carotenoid and can be classified as a precursor of vitamin A. Thus, it can be assumed that the antioxidant function of torularhodin has a theoretical basis in chemical structure.

In vitro cell model studies also help to establish the link between carotenoids and cytotoxicity [24]. In the present study, the effect of torularhodin on RAW 264.7 cells was investigated. In Figure 3C, the cytotoxic effect of 1.5 μM torularhodin was similar to that of 9 μM. The cytotoxic level of torularhodin was lowest at 6 μM (*p* < 0.05), indicating that torularhodin at specific doses contributes to cell proliferation.

### 2.4. Cellular Anti-Inflammatory Activity of Torularhodin in Sporidiobolus pararoseus ZQHL

It was shown that the TLR4 inflammatory pathway exacerbates cells’ inflammatory response by regulating the expression of a series of inflammatory genes [25]. To assess the molecular mechanisms by which torularhodin regulates the balance of cellular redox status, the activation of the TLR4 inflammatory pathway and the expression of downstream signaling factors were analyzed. The mRNA levels of TLR4, MyD88, TNF-a, IL-1β, COX-2, and iNOS were detected by qRT-PCR in each group, and the results are shown in Figure 4A–G. Compared with the Con group, the mRNA levels of inflammatory genes in the Mod group were elevated at different levels, while TLR4, IL-1β, COX-2, and iNOS transcripts were significantly elevated (*p* < 0.001). In addition, and MyD88 mRNA levels were also significantly increased (*p* < 0.01). In contrast, although inflammatory factor levels decreased in the Tor-L group, they were not more pronounced than in Tor-H, suggesting a dose effect of torularhodin’s inflammatory intervention. These results suggest that torularhodin reduces the expression of factors in the TLR4 inflammatory pathway in RAW 264.7 cells. Torularhodin has an immunomodulatory effect.

The TLR4 pathway has been regarded as a central link in the inflammatory response of macrophages, and physiologically, its downstream inflammatory factors, such as MyD88, play a significant role in regulating the immune response [26]. The association between inflammation was shown to be closely related to aberrant TNF-a expression [27]. On the one hand, the TLR4 pathway of LPS-induced RAW 264.7 cells was significantly activated, leading to a significant increase in the expression of pro-inflammatory cytokines and associated chemokines, including IL-1β, and TNF-a. On the other hand, upregulation of iNOS and COX-2 during inflammation indicated LPS-induced inflammation in RAW 264.7 cells, further suggesting that TLR4-activated signaling is involved in LPS-induced cellular inflammation. Furthermore, we found that torularhodin could protect against LPS-induced cellular inflammation by inhibiting pro-inflammatory cytokines.

## 3. Materials and Methods

### 3.1. Materials and Reagents

Lipopolysaccharides (LPS) (purity ≥ 99%) were purchased from Sigma, Dulbecco’s Modified Eagle medium (DMEM) was purchased from Cytiva, and penicillin-streptomycin antibiotics (1000 IU/mL penicillin and 1000 μg/mL streptomycin) were purchased from Gibco, and fetal bovine serum (FBS) was purchased from HyClone. Cell Counting Kit 8 (CCK-8) was purchased from Sigma. RNA extraction kit, cDNA reverse transcription kit, and SYBR qPCR reagent were purchased from Vazyme Biotech Co., Ltd. (Nanjing, China). The sequences of all primers were purchased from Shanghai Sangong (Shanghai, China). All other reagents were analytically pure.

### 3.2. Sample Collection and ITS Sequencing

The tested samples were taken from 6 vinegar enterprises, numbered a, b, c, d, e, and f. DNA extraction and PCR amplification were performed after sample collection. DNA was extracted from the samples using the Genomic DNA Purification Kit from Sangon Biotech Co., Ltd. (Shanghai, China). Fungal ITS RNA sequencing genes were amplified from whole genomes by primer pairs (fITS7, 5’-GTGARTCATCGAATCTTTG-3’; ITS4, 5’-TCCTCCGCTTATTGATATGC-3’). Amplification and sequencing were performed at 95 °C for 2 min, followed by 25 cycles of 95 °C for 30 s, 55 °C for 30 s, 72 °C for 30 s, and a final extension at 72 °C for 5 min. Subsequently, the PCR amplification products were detected by 2% agarose gel electrophoresis and were recovered and purified using the AxyPrep DNA Kit (Axygen Biosciences, Union City, CA, USA). PCR products were purified using QuantiFluor™-ST (Promega, Madison, WI, USA) for quantitative detection, qualified sequencing libraries were sequenced using the Novaseq PE250 (Illumina Inc., San Diego, CA, USA) platform, and each sample provided at least 50,000 pieces of Raw data. The DADA2 plugin of QIIME 2.0 was used to perform quality control, denoising, splicing, and chimera removal on the original fastq files to form amplicon sequencing variants. Finally, species annotation and visualization were performed using representative sequences and feature tables. The RDP classifier performed the analysis (http://rdp.cme.msu.edu/ (accessed on 14 August 2020)), and the fungal ITS rRNA was referenced to the UNITE database.

### 3.3. Isolation, Purification, and Cultivation of Fungi

The culture method of fungi refers to our previous research [7]. (1) The slant activation medium includes 20 g of glucose, 1 g of peptone, 1 g of yeast extract, and 1 L of water. (2) The liquid seed medium included 40 g of glucose, 20 g of corn-steep liquor, 1 g of KH_2_PO_4_, 0.5 g of MgSO_4_·7H_2_O, and 1 L of water. (3) The fermentation medium included 60 g of glucose, 10 g of steep corn liquor, 2 g of KH_2_PO_4_, 2 g of KH_2_PO_4_, 3 g of MgSO_4_·7H_2_O and 1 L of water, adding 0.1–0.3% (*v*/*v*) defoamer (25 mL). The feed medium included glucose at a concentration of 800 g/L and steep corn liquor at a concentration of 100 g/L. The final pH of the above medium was 6.0.

First, dilute the samples from the vinegar factory 100, 500, and 1000 times, respectively, spread them on the activated medium plate, and cultivate them at 30 °C until a single colony grows. The screen is according to the colony’s shape and color (orange-red), and select the bright red color and full-bodied colonies. Then, transfer the screened bacterial strains to the activation medium, cultivate them at 28 °C for 48 h until a single colony grows, and finally, select a single colony with the reddest color and inoculate it into the liquid seed medium, and culture it on a shaker at 28 °C 48 h.

Subsequently, the strain was inserted into the fermentation medium for cultivation, and the cultivation time was 64–72 h at 28 °C. After the bacterial liquid was obtained, ITS sequencing was performed. The sequencing results were BLASTed on NCBI for homology identification, and finally identified as lock-throwing Yeast, and sent to the China Center for Typical Microorganisms Collection (No. CCTCC NO: M 2021729) for preservation, named *Sporidiobolus pararoseus* ZQHL.

Finally, the strain was enriched and cultured in a fermenter, 35 L of fermentation medium was added to a 50 L fermenter, and the seed solution was added to the fermentation medium at an inoculum volume of 10% (*v*/*v*) and cultured at 28 °C for 64 h. The parameters during the fermentation process are shown in Appendix A; 800 g/L of glucose was added at 18 h of fermentation, and sugar was supplemented according to the residual sugar during 18–56 h to control the residual sugar in the fermentation system at 20 g/L and the dissolved oxygen at 20%.

### 3.4. Whole Genome Sequencing and Assembly

Whole genome sequencing was completed by Hanyu Bio-tech (Shanghai, China) [28]. First, the genome of *Sporidiobolus pararoseus* ZQHL was extracted by the SDS method, and the purity and integrity of the DNA were detected by agarose gel electrophoresis, quantified by Qubit. Then the genome-wide shotgun experiment was performed. Subsequently, different gDNA libraries (10–20 kb) were constructed and sequenced using Illumina Hiseq and third-generation PacBio sequencing technologies. Adapter sequences and reads with low sequencing quality values were removed. After filtering and trimming the reads, high-quality reads were assembled using SMRT Link v5.0.1 software to obtain a complete genome (https://www.pacb.com/support/software-downloads/ (accessed on 14 August 2020)).

### 3.5. Gene Function Annotation Analysis

Functional annotation of databases for coding gene sequences, identification of protein-coding regions, and gene prediction by combining ab initio gene prediction, transcriptome-based prediction, and homology-based prediction methods. Genome composition prediction used the Augustus 2.7 program to retrieve associated coding genes. Finally, the predicted genes were compared and annotated by BLAST with various functional databases, including GO (Gene Ontology) and KOG (euKaryotic Ortholog Groups).

### 3.6. Isolation and Purification of Torularhodin

After the fermentation broth was centrifuged at 5000× *g* for 10 min, the sludge was collected. The extraction method of torularhodin adopts the method of breaking the wall by acid-heating method [29]. First, the sludge and 3 M dilute hydrochloric acid were stirred and mixed at a ratio of 1:2 (*w*/*w*) and heated to 95 °C in a water bath for 5 min. Then, the samples were rapidly cooled in an ice-water bath and centrifuged at 5000× *g* for 20 min to obtain the yeast sludge after breaking the wall. Subsequently, the supernatant was discarded, 200 mL of absolute ethanol was added to 1 kg of yeast sludge, stirred thoroughly for 10 min, and then 1000 mL of n-hexane was added for stirring. Finally, the upper organic phase was collected, and the organic phase was recovered by rotary evaporation at 40 °C. The residue was oil rich in carotenoids. The separation, purification, and quantification of torularhodin refer to the previous method. Silica gel column, HPLC, and LC-MS were used to separate and purify torularhodin. Torularhodin was obtained and identified by the HPLC method (>96% purity) [30,31].

### 3.7. Free Radical Scavenging Ability of Torularhodin In Vitro

In ABTS (2,2’-azino-bis(3-ethylbenzothiazoline-6-sulfonic acid)* radical, first, ABTS radical cation (ABTS*) solution was prepared by mixing 7 mM ABTS solution with an equal amount of potassium persulfate. The obtained mixture was mixed and stored in the dark at 4 °C for 15 h. After the ABTS* solution was diluted with absolute ethanol, the absorbance at 30 °C and 734 nm was 0.70. Then, the sample was mixed with the ABTS* solution (1:9) and reacted for 10 min, measuring the absorbance at 30 °C and 734 nm. The DPPH free radical scavenging experiment is to mix the samples at different concentrations with DPPH (0.04 mg/mL) diluted with methanol. After the samples were reacted for 30 min, the absorbance was measured at 517 nm at room temperature in the dark. The free radical scavenging activity formula is as follows:ABTS*free radical scavenging activity (%) = 100 − [(Aa − Ab) × 100/Ac]
DPPH free radical scavenging activity (%) = 100 − [(Aa − Ab) × 100/Ac]

Note: Aa is the absorbance of the sample, Ac is the absorbance of the blank, and Ab is the absorbance of Ab, *n* = 6.

### 3.8. Effect of Torularhodin on LPS-Induced Inflammation in RAW 264.7 Cells

The cellular inflammation induction model refers to previous research [32], RAW 264.7 mouse macrophages were cultured in DMEM containing fetal bovine serum, penicillin, and streptomycin (37 °C, 5% CO_2_). Change the medium every 1–2 days and pass the passage every 2 days. When the cells grew to 80–90% confluence, the cells were subcultured at a ratio of 1/3. Torularhodin was dissolved in DMSO and diluted in different concentrations with DMSO.

First, cytotoxicity was tested by measuring cell viability using the CCK-8 assay. First, 100 uL of culture medium containing cells was inoculated into a 96-well plate (10,000 cells/well), and after 24 h of culture, samples of different concentrations were added to continue culture for 24 h. Subsequently, 100 uL of CCK-8 working solution (CCK-8/DMEM, 1:10) was added and incubated in the dark for 60 min. Measure the absorbance at 450 nm using a microplate reader. Cell survival rate = [(experimental well − blank well) /(control well − blank well)] × 100%. In addition, in the LPS cell inflammation model experiment, the medium containing cells was first inoculated into a 24-well plate (1,000,000 cells/well). After 24 h of culture, LPS (the final concentration of the system was 100 ng/mL, dissolved in DMEM) and samples (the final concentration of the system was 6 and 10 μM, dissolved in DMSO) were added. Furthermore, it continued to incubate for 24 h. Cell wells were divided into control group (Con), model group (Mod, LPS), torularhodin low-concentration group (Tor-L, LPS plus torularhodin, the final concentration of the system of torularhodin was 3 μM), torularhodin high-concentration group (Tor-H, LPS plus torularhodin, the final concentration of the system of torularhodin was 6 μM).

### 3.9. Real-Time Quantitative PCR Analysis

Cells were harvested after trypsinization, and total mRNA was extracted according to the manufacturer’s protocol. Their concentrations were measured using a BioDrop Duo. The absorbance ratio at 260/280 nm between 1.8 and 2.0 was acceptable and diluted to the same concentration. Subsequently, the total mRNA was reverse-transcribed into cDNA using a cDNA synthesis kit on a thermal cycler. Finally, RT-qPCR was performed on the StepOnePlus Real-Time PCR System. β-action was used as a reference gene for normalization. The relative expression of genes was calculated according to the 2^−ΔΔCt^ method. Primer sequences were referred to in Appendix A.

### 3.10. Statistical Analysis

Statistical analysis was performed using GraphPad Prism 7. Multiple group data comparisons were performed using one-way analysis of variance (ANOVA), with Duncan’s correction for post hoc testing. *p* * < 0.05, *p* ** < 0.01 and *p* *** < 0.001 indicated statistical significance.

## 4. Conclusions

In conclusion, our study showed that *Sporidiobolus pararoseus*, named *Sporidiobolus pararoseus* ZQHL, was isolated and screened from vinegar samples. *Sporidiobolus pararoseus* ZQHL has a safe and natural origin. The torularhodin produced by *S. pararoseus* ZQHL exhibited significant antioxidant activity in in vitro DPPH and ABTS scavenging assays and improved cellular activity. Furthermore, in a model of LPS-induced inflammation in RAW 264.7 cells, torularhodin treatment effectively attenuated LPS-induced cellular inflammatory damage in a dose-dependent manner. It also reduced the expression of inflammatory factors such as TLR4, MyD88, and TNF-a in cells by a mechanism that attenuated the cellular inflammatory response through inhibition of the TLR4 inflammatory pathway. Thus, in vinegar samples, torularhodin produced by *S. pararoseus* fungi significantly scavenged free radicals in vitro and alleviated RAW 264.7 cellular inflammation by modulating the TLR4 pathway.

## Figures and Tables

**Figure 1 molecules-28-01436-f001:**
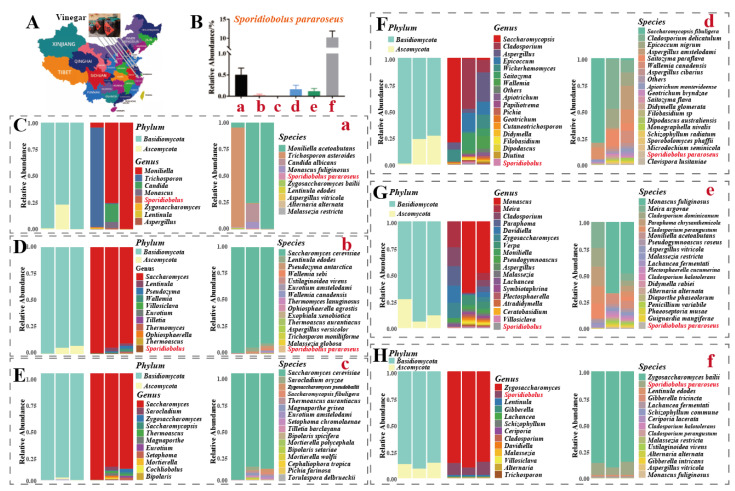
Source and distribution of *Sporidiobolus pararoseus* in vinegar. (**A**) Sampling map of vinegar samples. (**B**) Relative abundance of *S. pararoseus* in vinegar samples. (**C**–**H**) ITS sequencing results of six vinegar samples.

**Figure 2 molecules-28-01436-f002:**
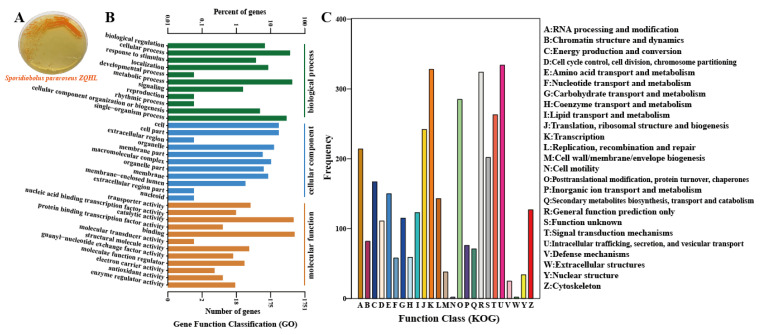
Culture morphology and whole genome sequencing of *Sporidiobolus pararoseus* ZQHL. (**A**) Culture morphology of *S. pararoseus* ZQHL. (**B**) Functional annotation of GO of *S. pararoseus* ZQHL. (**C**) Functional annotation of the COG of *S. pararoseus* ZQHL.

**Figure 3 molecules-28-01436-f003:**
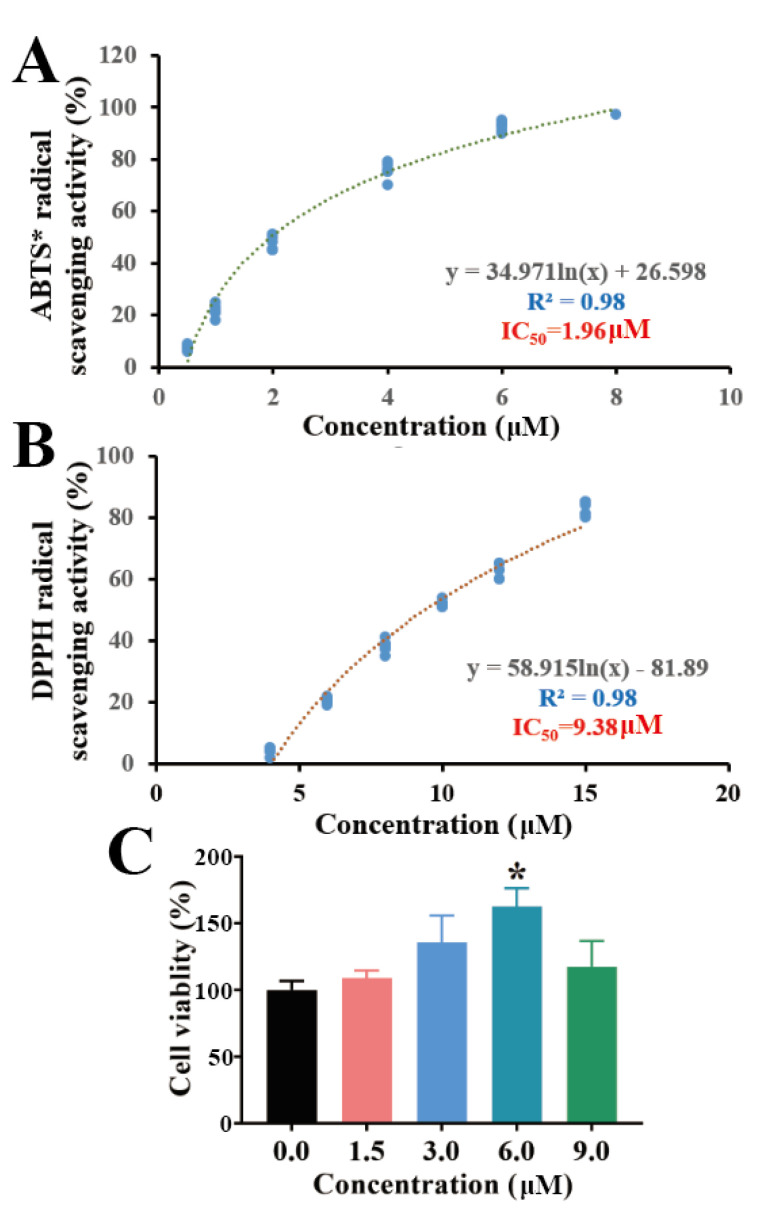
In vitro antioxidant capacity and cytotoxicity of Torularhodin. (**A**) Effect of Torularhodin on the in vitro antioxidant activity of ABTS. (**B**) Effect of Torularhodin on the in vitro antioxidant activity of DPPH. (**C**) Effect of Torularhodin on the cytotoxicity of RAW 264.7. Logarithmic equation in logistic regression was used in regression equation. * *p* < 0.05 indicated statistical significance compared with “0.0” group.

**Figure 4 molecules-28-01436-f004:**
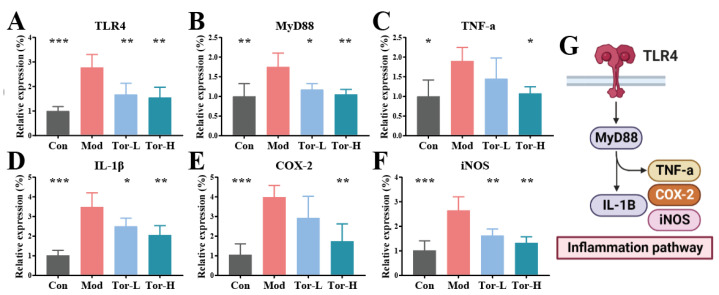
Effect of torularhodin on the RAW 264.7 inflammatory cell model. (**A**) Relative expression levels of TLR4. (**B**) Relative expression levels of MyD88. (**C**) Relative expression levels of TNF-a. (**D**) Relative expression levels of IL-1β. (**E**) Relative expression levels of COX-2. (**F**) Relative expression levels of iNOS. (**G**) TLR4 inflammatory pathway. Significant differences between Mod group and other groups: * *p* < 0.05, ** *p* < 0.01, *** *p* < 0.001.

## Data Availability

Not applicable.

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
