# Peer review of "Study on the Cellular Anti-Inflammatory Effect of Torularhodin Produced by Sporidiobolus pararoseus ZQHL Isolated from Vinegar Fungus"

_molecules, 2023, doi:10.3390/molecules28031436_

Round 1

Reviewer 1 Report

The study deals with anti-inflammatory effect of the fungal carotinoid Torularhodin. Brief literature search revealed several papers that report anti-inflammatory activity of this compound. Please cite them appropriately, e.g. https://www.mdpi.com/1420-3049/27/19/6398; https://pubs.acs.org/doi/abs/10.1021/acs.jafc.9b03847; https://pubs.acs.org/doi/abs/10.1021/acs.jafc.0c01892.

Correct or explain the following:

Line 274: "1.96 in ABTS μM".

Line 280: peroid is missing.

Line 282: "on the cytotoxicity of RAW".

Figure 3A, 3B: please explain the choice of regression method. It seems inappropriate, likely sigmoidal regression should be used.

Figure 3C: cell viability range from 0 to 2% on y-axis does not make sense.

Figure 3 and elsewhere: greek "micro" should be used instead of "u" in micromoles.

Please specify explicitly what exactly Tor-L and Tor-H mean, what is the concentration (not dose, since it is used in vitro).

Isolation and purifiaction of Torularhodin should be a separate section in the Material and section, not a part of section 2.6.

Purity and identity of Torularhodin should be confirmed in SI, please provide the respective HPLC and LC-MS data.

Author Response

Please see the pdf file. Thanks!

Reviewer 2 Report

Study on the cellular anti-inflammatory effect of torularhodin produced by Sporidiobolus pararoseus ZQHL isolated from vinegar fungus

C. Liu, M. Han, F. Lv, Y. Gao, X. Wang, X. Zhang, Y. Guo, Y. Cheng, H. Qian

I would congratulate the authors for their excellent manuscript. The manuscript, as written, is clear to understand. The authors have sequenced one colony of Sporidiobolus pararoseus. The selection was based on a high presence of Torularhodin. a carotenoid with interesting anti-inflammatory (anti-oxidant) properties. The colony was named Sporidiobolus pararoseus ZQHL, and the authors studied the properties of Torularhodin on the RAW 264.7 inflammatory cell model. 

I only have one central question related to the study of the cytotoxic effect of torularhodin on RAW 264.7 cells (Line 281). Does this study was performed on RAW 264.7 cells, which have been undergoing inflammation via LPS induction? Or were the RAW 264.7 cells used for this study untreated?

Minor suggestions: 

To help the average reader, I suggest the authors to

1.     Add the formula for cell viability in the legend of Figure 3C. In particular, the scale in the graph: % (x100).

2.     The supplementary section should include the HPLC and LC-MS results (raw data) for torularhodin's separation, purification, and quantification.

Author Response

Please see the pdf file. Thanks!

Round 2

Reviewer 1 Report

Please correct the following:

line 209: the final concentration of the system of torularhodin was torularhodin

line 290: the effect of torularhodin RAW 264.7 cells

Author Response

Dear reviewer:

  Thanks very much for considering our presented manuscript and helping us to improve the scientific and linguistic quality of the manuscript. Corresponding revisions have been made according to the your respected suggestions and comments. We hope that the revisions and the revised manuscript now are acceptable for approval. 

Comment 1 

line 209: the final concentration of the system of torularhodin was torularhodin

Author reply:

Thanks very much for your conducive suggestion. We have checked and revised the sentence: "the final concentration of the system of torularhodin was 6 μM".  Corresponding changes were marked with red color in the revised manuscript. (Line 209).

Comment 2

line 290: the effect of torularhodin RAW 264.7 cells

Author reply:

Thanks very much for your respected comment. We have checked and revised the sentence: "In the present study, the effect of torularhodin on RAW 264.7 cells was investigated".  Corresponding changes were marked with red color in the revised manuscript. (Line 290).